# Comprehensive feature evaluation of the main facades of Catholic churches in Sichuan-Chongqing region based on semantic difference method, 1840–1949

Qin Lai[1], Xinkai Li[2], Beibei Zhang[2], Nan Chen[1], YunZhang Li[1]*

1 College of Architecture and Environment, Sichuan University, Chengdu, China, 2 Architecture Studio, Sichuan Provincial Architectural Design and Research Institute Co., Ltd., Chengdu, China

* liyunzhang@scu.edu.cn

## Abstract

The existing Catholic churches constructed in the Sichuan-Chongqing region between 1840 and 1949, as precious carriers of Sino-Western cultural integration, have been listed in the cultural heritage protection catalog. This study employs the semantic difference method to conduct a quantitative evaluation of the main facade characteristics of these churches across five dimensions, operationalized through six specific evaluation criteria: relationship with the environment, facade contour, facade style 1 (Chinese vs. Western), facade style 2 (Folk vs. Official), facade decoration, and religious expression. The correlation between these facade features was also analyzed. The findings reveal: (1) Among the 62 existing Catholic churches in Sichuan and Chongqing, 14 are Chinese-style churches, accounting for 22.6%. (2) Facades demonstrate distinct cultural fusion: Chinese-style churches incorporate Western vertical composition elements, while Western-dominated churches integrate environmental harmony, Chinese architectural style, folk styles, and minimalist decoration. (3) According to the correlation analysis of Spearman's correlation coefficient, the correlation coefficient between the facade style and decorative characteristics of the church is 0.857, and the correlation coefficient with religious expression is 0.754, showing a significant positive correlation. This study aims to provide a reference for the quantitative evaluation of cultural heritage facade characteristics and preservation of Sino-Western architectural cultural integration.

## Introduction

Between 1640 and 1949, Catholicism saw significant church construction in the Sichuan-Chongqing region [1], particularly after the Qing Dynasty's forced lifting of the Catholic ban following the 1842 Opium War. The 1890 Treaty of Yantai's Supplementary Provisions granted Chongqing commercial port status, enabling Western

**Data availability statement:** All relevant data are within the manuscript and its Supporting information files.

**Funding:** The author(s) received no specific funding for this work.

**Competing interests:** The authors have declared that no competing interests exist.

missionaries to launch large-scale missionary activities in Sichuan and Chongqing [2]. Statistics from the *Catalogus Ecclesiae Catholicae in China (1935–1936)* indicate that Sichuan then hosted 8 vicariates and prefectures, comprising 234 cathedrals and churches and 414 chapels [3]. These surviving Catholic churches stand as living witnesses to Sino-Western cultural integration [4], with their architectural facades reflecting historical development. Consequently, a systematic study of these church facades in the Sichuan-Chongqing region holds immense historical and cultural significance [5].

Since the publication of China's "Three-Self Declaration" in 1950, which promoted the church to achieve "self-governance, self-support, and self-propagation," it marked the beginning of the Sinicization of Christianity in China. It actively facilitated the adaptation of various aspects of Chinese Christianity, such as theological doctrines, worship practices, organizational structures, and missionary methods, to suit China's national conditions, public sentiments, and the psychological acceptance and spiritual expression of the Chinese people [6]. Although the church facades built before 1950 in this study predominantly reflect Western religious architectural concepts, they encapsulate significant stylistic evolutions. According to Coomans, Catholic architecture in China underwent two major paradigm shifts: a transition toward Western "triumphalist" styles, such as Gothic, after 1900, followed by a pivot toward localized and modern architectural forms in the late 1920s [3]. Despite their historical significance, these pre-1950 structures are progressively dwindling in number [7]. As designated cultural heritage sites, they represent irreplaceable architectural treasures. Since the facade serves as the primary medium for expressing religious doctrine, a systematic study into these features is essential for the authentic preservation and restoration of this unique cultural legacy.

As McDannell [8] observes, churches emphasizing etiquette carefully consider how visual art enhances or hinders worship. The facades of Catholic churches in the Sichuan-Chongqing region possess exceptional artistic and cultural value [9], with their main facade serving as the central stage for showcasing artistic and cultural expressions. However, current research on church facades in this region predominantly focuses on manifest stylistic elements and decorative features [10], while comprehensive studies of their holistic characteristics remain insufficient. Particularly, research on the implicit aspects of facades shows considerable potential for exploration. Although Trisno [11] noted the sacred symbolic significance of church facades, quantitative analysis of this symbolism remains underdeveloped. Moreover, existing studies primarily employ qualitative approaches, lacking quantitative evaluation methods and comprehensive assessments of facades' multifaceted attributes.

Catholic churches in the Sichuan-Chongqing region are religious buildings introduced to China from the West, embodying their religious spirit. After their introduction to China, church architecture underwent a process of localization [3]. In different periods, the church sought to demonstrate its influence or make Western religious architecture more acceptable to local residents [12]. The construction of church facades fully considered the psychological activities of local residents, expressing Western Catholic doctrines either directly or subtly from the inside out [13]. Therefore,

this study introduces the semantic difference method (SD method) based on psychological judgment to construct evaluation factors and select corresponding adjectives. It quantitatively assesses the comprehensive and typical characteristics of the main facades of Catholic churches in the Sichuan-Chongqing region from different dimensions, considering both Chinese and Western tendencies. Additionally, by calculating the average evaluation score of each church facade, it analyzes the localization characteristics and the integration of Chinese and Western elements at the individual level of Catholic churches with varying degrees of Chinese and Western influences in the Sichuan-Chongqing region. Furthermore, the relationship between facade characteristics are explored by analyzing both the overall results of the entire sample group and the detailed evaluation scores of specific representative churches. This provides new insights into the inheritance of architectural cultural heritage and helps people recognize the uniqueness of Catholic churches in the Sichuan-Chongqing region. The research question (RQ) is as follows:

RQ1: What are the overall and typical features of the main facade of Catholic churches in the Sichuan-Chongqing region?

RQ2: What are the localization characteristics and ways of blending Chinese and Western elements in different types of Catholic churches in the Sichuan-Chongqing region?

RQ3: What are the interrelationships and associations among the facade characteristics of Catholic churches in the Sichuan-Chongqing region?

The structure of the rest of this paper is as follows: Section 2 introduces the review of relevant literature; Section 3 describes the research methods; Sections 4–5 present and discuss the research results; and Section 6 provides the conclusions.

## Literature review

### Church facade

Current research on the characteristics of church facades can be divided from a visual perspective into studies of explicit facade features and implicit features. The study of explicit facade features mainly focuses on parameter design, style, and decoration. In the research on facade parameter design, some scholars have found that the exterior facade of St. Dominic's Church in Macau exhibits a decreasing upward trend in terms of hierarchy, modularity, and the golden ratio [14]; others proposed using finite element and discrete macro-element models to analyze the exterior wall parameters of St. Mary Magdalene's church [15]. In the study of church facade style features, the window openings of the Hongjialou Catholic church in Jinan, China, are decorated with stone strip patterns arranged in an outward-facing figure-eight shape. This design features a typical Gothic rose window, which is recognized as a Western Gothic style [16]. Another scholar used the Catholic church in Anqing as a typical case to systematically analyze its architectural translation strategies and the logic of its integration mechanism generation [4]. More detailed research involves the decorative features of facades, with some scholars analyzing the color schemes of architectural structures in Baroque architecture [17]. Some scholars used 3D laser scanning technology and geographic information system technology to determine the rock types of the facade of St. Nicholas church [18]. Regarding the implicit characteristics of church facades, scholars have identified symbolic elements and sacred representations that influence the architectural form of Catholic churches [19]. Others have analyzed the overall architectural form and decorations of churches, noting that the form of Catholic churches embodies divinity and symbolic imagery. This architectural mass is larger than other components, and small spires serve as decorative elements that create sacred value [11]. Additionally, researchers have explored the connections between church facades and other elements by investigating the variation patterns of reverberation time in Italian Catholic churches across architectural styles and spatial volumes [20]. Studies have also examined the correlation between the facade designs of seven Fieschi-style churches and changes in the frequency and intensity of slave raids [21].

Research on churches in the Sichuan-Chongqing region has been conducted at various scales, including Southwest China, Sichuan-Chongqing, Chongqing, Southern Sichuan, and Western Sichuan, yielding relatively abundant results.

The research content not only includes the spatial distribution and driving factors of churches at the macro level [1,22], but also the plan layout of church complexes at the meso level [23]. In the micro-level studies of individual church facades related to this study, scholars have analyzed the Sino-Western style characteristics of churches in the Sichuan-Chongqing region, pointing out that most Catholic churches in Chongqing exhibit basic features of Western classical architecture, including Gothic, Romanesque, Baroque, Rococo, and Sino-Western hybrid styles [24]. The architectural style characteristics of Catholic churches in Southern Sichuan include Gothic features, Baroque features, regional traditional styles, and the mutual penetration of Sino-Western architectural styles [25,26]. Some scholars have focused on the Chinese characteristics of churches, noting that the facade designs of Western Sichuan Catholic churches largely inherit local styles while integrating Western decorative elements in traditional forms [27], and have analyzed the Sinicization of Catholic churches in the Southwest region in terms of external morphology and architectural decoration [28,29]. Some scholars argue that the facades of Catholic churches in the Sichuan-Chongqing region exhibit diverse forms, with distinct characteristics blending Chinese and Western elements. These facades typically combine two or more styles, making it difficult to categorize them into a single architectural style. Therefore, they are classified into two distinct dimensions based on stylistic differences: those with pronounced Western features and those with pronounced Chinese features [30].

In summary, while extensive research has been conducted on the visible characteristics of churches, studies focusing on their hidden features remain insufficient. Numerous qualitative analyses have been performed on the facade styles of Catholic churches in the Sichuan-Chongqing region, revealing that these churches exhibit multiple Western and indigenous architectural styles. However, such studies have primarily focused on stylistic and decorative dimensions. There is a notable lack of comprehensive evaluations of Catholic church facades in the region, particularly those that systematically assess their integrated characteristics from external to internal aspects and from macro to micro perspectives.

## Semantic difference method

The semantic difference method is a quantitative approach that measures people's subjective perceptions and attitudes toward objects by combining linguistic or perceptual symbols with antonymous adjective scales [31]. This method quantifies psychological responses and has been applied in various fields: categorizing perfume bottles by shape [32], assessing user satisfaction with historical district renovations [33], evaluating automotive instrument panels [34], and testing perceptions of church facades [35]. Meanwhile, establishing evaluation criteria enables comprehensive assessment of objects, thereby identifying key influencing factors. For instance, when evaluating the waterfront landscape of the Zhonghuamen section of the Qinhuai River, we developed assessment parameters from three dimensions: landscape design, ecological sustainability, and cultural heritage. The analysis revealed that users consistently rated the area's water accessibility, shoreline morphology, aesthetic appeal, environmental cleanliness, and historical continuity as highly satisfactory [36]. Similarly, analyzing tourists' impressions of classical garden rockeries identified four key factors: environmental harmony, rockery design, conceptual intent, and stacking techniques [37]. This methodology enables comparative analysis of factor weights to identify optimal configurations. For instance, evaluations of therapeutic effects in campus courtyard designs revealed that plants positioned at or above human eye level demonstrate superior healing efficacy compared to grassy lawns. Irregular plant arrangements exhibit greater aesthetic appeal than monotonous layouts while effectively diverting visual focus [38]. Similarly, assessments of Hefei's traditional commercial districts identified street corridors, horse - head walls, and landmark features as the most visually striking elements [39]. Regarding the enhancement of architectural integration in historical urban landscapes, research indicates that visual quality depends on coherence rather than dramatic impact [40].

In summary, the SD method can classify objects by quantifying psychological perceptions, establish a scoring mechanism to obtain evaluation results, and analyze the influencing factors and their weights. In cross-cultural comparisons, most comparative studies become extremely challenging when dealing with intangible attributes [31]. As a product of Sino-Western cultural integration, Catholic churches in the Sichuan-Chongqing region currently rely predominantly on

subjective impressions rather than quantitative analysis for facade evaluations. Therefore, this study employs the SD method to construct evaluation factors and select adjectives, conducting a comprehensive assessment of Sino-Western characteristics in the facades of Catholic churches in the Sichuan-Chongqing region.

## Research methodology

### Research area

This study focuses on the existing Catholic churches built in the Sichuan-Chongqing region from 1840 to 1949. Before 1949, Sichuan and Chongqing were part of the same provincial administrative region [41], so this study treats the Catholic churches in Sichuan and Chongqing as a whole. The Sichuan-Chongqing region is located in the southwest of China, with complex topography and geomorphology [42], which makes the Catholic churches in this region highly distinctive. Based on systematic data collection from the Digital Local Chronicles of Sichuan and Chongqing and historical records in *Sichuan Catholicism* [43], 62 surviving Catholic churches built within this timeframe were identified. The 62 existing Catholic churches in the Sichuan-Chongqing region are mainly distributed in the central [1], eastern, and southwestern parts of the region. The central area has a relatively dense distribution of churches, while the eastern and southwestern areas are relatively sparsely distributed (Fig 1). Given that the main facade of the chapel is the core area of the church complex expressing religious doctrines, this study selects it as the research object. The forms of the churches are diverse, showing obvious tendencies of Chinese and Western styles.

### Research approach and research design

This study employed the SD method to evaluate the main facade characteristics of 62 Catholic churches in the Sichuan-Chongqing region. The research process was divided into three phases: preparation, investigation, and analysis. During the preparation phase, we photographed and selected images of the main facade of each of the 62 churches along with their environmental backgrounds, while assigning unique identifiers to each structure. Evaluation criteria were established based on the intrinsic meanings of facade features and the predominant Chinese-Western architectural tendencies. For each criterion, we identified easily understandable, clearly defined, and contrasting adjectives and designed a graded

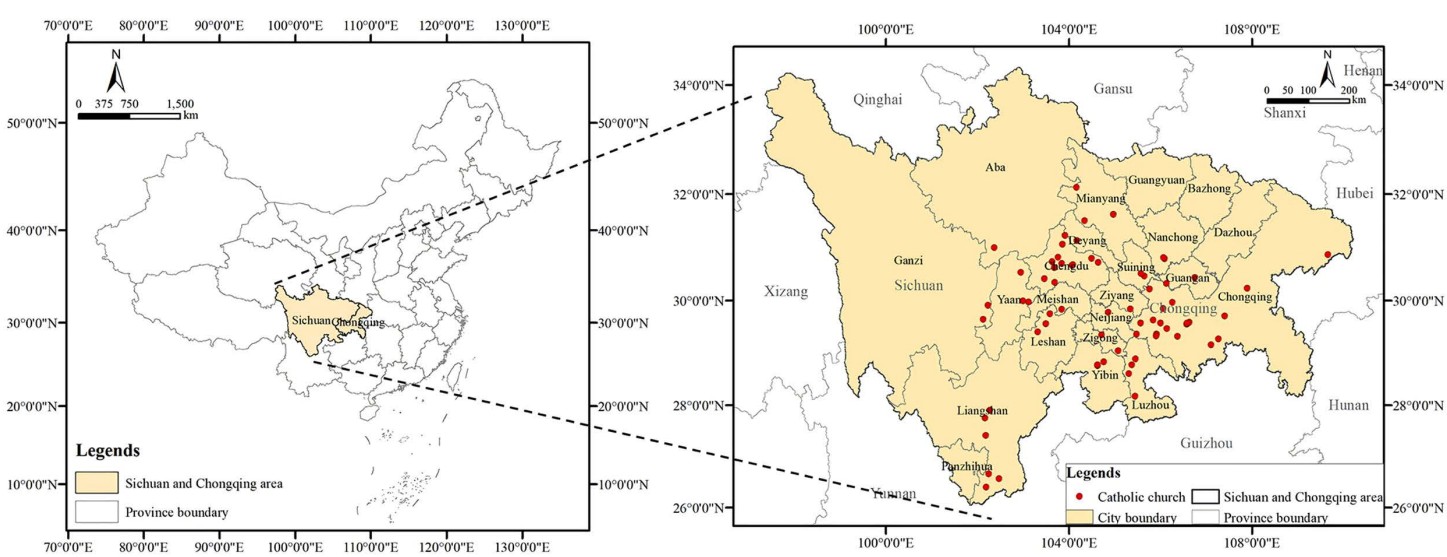

**Fig 1. Distribution of Catholic churches in Sichuan and Chongqing.** (Source: Drawn by the authors based on field surveys and *Sichuan and Chongqing Digital Local Chronicles*)..

evaluation scale. In the investigation phase, participants were invited to view the facade images of all 62 churches and complete the graded evaluation scale. The collected scoring data were then statistically analyzed. The analysis phase began with reliability and validity verification of the data. After confirming data validity, descriptive statistical analysis was conducted on the SD evaluation scale to identify comprehensive characteristics of church facades at the regional level, as well as typical features and localized characteristics of Chinese-Western architectural integration at the individual building level. Finally, correlation analysis was performed between evaluation factors to explore the potential associations and patterns among different facade elements.

**Experimental design.** Scholars evaluate facade characteristics across multiple dimensions, primarily focusing on form, style, ornamentation, materials, and color [44]. Emotional elements also form a fundamental component in architectural facade assessment [45]. From a broader perspective, the harmonious relationship between architectural facades and their environments constitutes another facet of facade characteristics [46]. Through scholarly definitions of facade features, on-site observations of the church, priestly interpretations, and expert discussions, the following five dimensions were identified as evaluation factors for assessing the facade characteristics of Catholic churches: the relationship between the facade and its environment, the contour characteristics of the facade, stylistic features, decorative features, and religious expression.

Furthermore, through analyzing the oriental-western influences in the facades of 62 churches, we identified evaluative adjectives across five dimensions and six specific evaluation criteria that demonstrate both positive and negative correlations, reflecting the interplay between Chinese and Western architectural tendencies. In the dimension of the relationship between the facade and the environment, Charles Borromeo, a Catholic saint venerated by the Church, proposed in his "Manual of Church Architecture and Decoration" that churches should be situated in prominent locations [47]. Consequently, the positive adjective is "integrated" while the negative one is "prominent." Regarding the dimension of facade contour characteristics, as sacred architecture embodying divine concepts, churches should adopt a "tapering upward" compositional form to enhance verticality [19,48], resulting in the positive adjective "horizontal" and the negative one "vertical." In terms of facade stylistic features, the facades of Catholic churches in the Sichuan-Chongqing region exhibit diverse forms with distinct Chinese-Western hybrid characteristics. They typically blend two or multiple styles, making it challenging to categorize them into a specific style. Based on stylistic differences, they can be classified into two dimensions: distinctly Western features and distinctly Chinese features [25,30]. Furthermore, the church facade exemplifies a personalized interpretation of localism, where official architectural styles are blended with "folk elements" [49]. This is reflected in the antonymous adjectives: "Chinese-style" versus "Western-style," and "folk" versus "official." In terms of decorative features, the church employs exquisite stone carvings and intricate ornamentation, striving for perfection [50], contrasting with the antonyms "simple" versus "complex." Regarding religious doctrine, the Catholic Church views the visual representation of the church as a tool for religious education [51], with antonyms being "direct" versus "implicit."

The evaluation scale is designed as a 9-point scale, where 1 indicates adjectives with a positive leaning, more characteristic of Chinese culture, while 9 indicates those with a negative leaning, more characteristic of Western culture. The number 5 represents a neutral midpoint, signifying a balance between Chinese and Western characteristics. To ensure professional rigor in the evaluation, a purposive sampling method was employed to select 50 architecture students as participants [52]. The inclusion criteria required participants to be senior undergraduate or graduate students who had completed courses in Architectural History and Architectural Design, ensuring they possessed the necessary aesthetic and professional expertise to evaluate church facades. The study utilized 62 church facade photographs, which were sequentially displayed in a classroom as numbered slides. Participants completed rating scales to evaluate each church. After the survey, the completed forms were collected and the data were systematically organized.

**Analytical framework.** The analysis phase comprises three steps (Fig 2). The first step conducts reliability testing on collected data from the evaluation scale using SPSS software, including Cronbach's Alpha coefficient for reliability assessment, KMO adequacy test, and Bartlett's test of sphericity. The Cronbach's Alpha coefficient evaluates whether

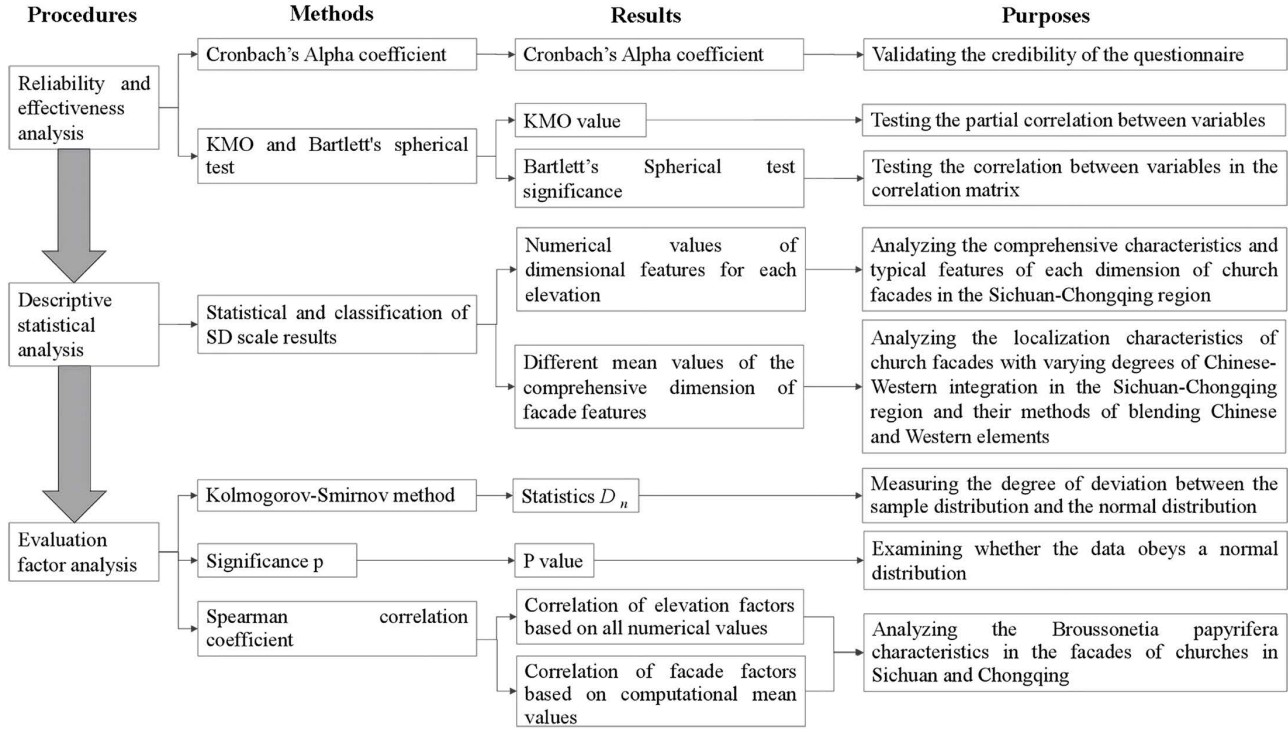

**Fig 2. Analysis framework.**

multiple items in the scale measure the same construct (S1 Appendix Formula 1). The KMO test assesses the suitability of data for factor analysis (S1 Appendix Formula 2), while the Bartlett's test of sphericity determines sufficient correlations between variables. A Cronbach's Alpha coefficient $\alpha > 0.8$ indicates excellent reliability, $0.7 < \alpha \leq 0.8$ indicates good reliability, $0.6 < \alpha \leq 0.7$ indicates acceptable reliability, and $\alpha \leq 0.6$ requires scale modification. A $KMO > 0.8$ indicates suitability for factor analysis, $0.7 < KMO \leq 0.8$ indicates suitability, $0.6 < KMO \leq 0.7$ indicates marginal suitability, and $KMO \leq 0.6$ indicates unsuitability. The Bartlett's test of sphericity is determined by the significance P-value: if $P < 0.05$, items show significant correlations and are suitable for factor analysis; if $P \geq 0.05$, factor analysis is unsuitable.

The second step involves analyzing the scores of each evaluation factor in churches using descriptive statistics from the SD evaluation scale. This analysis aims to preliminarily characterize the comprehensive features of church facades in the Sichuan-Chongqing region and their tendencies toward Chinese-Western integration. Additionally, by categorizing the evaluation factors into different score ranges, the study examines the localization characteristics and fusion approaches of Chinese-Western architectural styles in churches across various integration levels within the region.

The third step analyzes the interrelationships among individual facade factors at the architectural level to reveal structural characteristics. This study conducted a correlation analysis using all evaluation scores collected through the assessment scale and calculated the mean data. First, the Kolmogorov-Smirnov (K-S) test was employed for large samples ($n > 50$) to evaluate normality, with the Dn statistic calculated as shown in S1 Appendix Formula 3−4. Subsequently, significance p-value tests were performed using the formula in S1 Appendix Formula 5 to verify normal distribution. Spearman's correlation analysis was then applied to examine relationships between six factors, with the $\rho$ coefficient calculated as shown in S1 Appendix Formula 6. The Dn statistic ranges from [−1, 1], where higher values indicate greater deviation. A p-value $< 0.05$ indicates a non-normal distribution, while p-value $\geq 0.05$ confirms a normal distribution. Spearman's

coefficient values range from [−1, 1]: ρ = 1 signifies perfect positive correlation, ρ = −1 indicates perfect negative correlation, and ρ = 0 denotes no correlation.

## Declaration

The recruitment period for this study was from March 6, 2025 to March 7, 2025. Participants strongly support and trust this research and have obtained informed consent for its publication. All the participants were adults. Since the study focused on the visual evaluation of publicly accessible church façades and involved a non-invasive questionnaire survey, no specific field site permits were required. The survey was conducted in accordance with the Declaration of Helsinki. All student participants were informed of the study's purpose, and written informed consent was obtained prior to their participation.

## Data analysis

### Data reliability test results

This study analyzed factor scores from 62 churches, with 50 valid responses collected through evaluation scales, yielding 3,100 data points. The Cronbach's Alpha coefficient ranged between 0.7 and 0.88 (Fig 3), indicating strong internal consistency and high reliability in the facade composite characteristics. The KMO value also fell within this range, suggesting strong partial correlations among variables that reveal the intrinsic structure of church facades. Furthermore, the Bartlett's test results showed p-values all below 0.001, which further confirmed the suitability of the collected data for factor analysis. The Cronbach's Alpha coefficient, KMO value, and significance value not only validate the reliability and validity of the data, but also directly support the scientific validity of the core concept of church facade comprehensive characteristics.

### Numerical analysis of facade dimensions in Sichuan-Chongqing Catholic churches

Based on the validity of the data, this section analyzes the comprehensive and typical characteristics of the relationship between the church facade and the environment, the facade contour, the facade style, the facade decoration and the

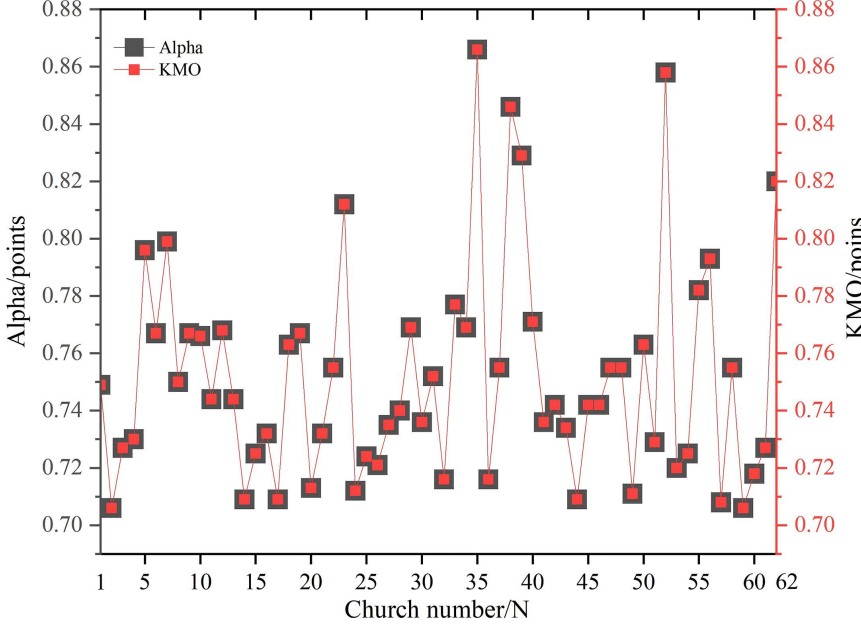

**Fig 3. Reliability test results.**

religious expression of the church facade from the regional level according to the scores of the survey objects on the facade characteristics of the church in Sichuan and Chongqing.

**Characteristics of the relationship between church facade and environment.** Statistical analysis of 62 churches in the Sichuan-Chongqing region shows that scores are primarily concentrated in two ranges: 5–8 points and 2–4 points (Fig 4a). Overall, there are 41 churches with a score above 5, accounting for 66.1%, and 21 churches with a score below 5, accounting for 33.8%. This indicates that more Catholic churches in the region stand out from their surroundings, with fewer blending into the environment. The average score of the relationship between churches and environment in Sichuan and Chongqing region is 5.34, which shows a prominent feature on the whole.

Among these churches, the highest environmental compatibility score was 7.94, with a median of 5.9 and a minimum of 1.88. Analysis of the relationship between churches and their surroundings through comparison of maximum, median, and minimum values reveals distinct patterns: High-scoring churches feature tall towers whose height and architectural form create a sense of disconnection from the environment. Medium-scoring churches achieve harmony by reducing their tower height to match surrounding residential buildings, creating a balance of prominence and integration. Low-scoring churches adopt forms resembling nearby dwellings, with their height aligned with tree planting heights, seamlessly blending into the environment.

**Characteristics of church facade contour.** Statistical analysis of architectural profile scores from 62 churches in the Sichuan-Chongqing region reveals that most churches fall into two score ranges: 6–8 points and 1–4 points (Fig 4b), demonstrating a predominantly Western-style vertical development pattern. Among them, there are 47 churches with a score of 5 or above, accounting for 75.8%, and their architectural outline tends to be vertical; while there are 15 churches with a score below 5, accounting for 24.2%, and their architectural contour tends to be horizontal. The average contour score was 5.75 points, indicating a significant numerical advantage of vertical-profile churches over horizontal-profile ones.

In these churches, the facade contour scores range from a minimum of 1.48 to a maximum of 8.24 points, with a median of 6.38. Analysis of the three key metrics reveals distinct characteristics: High-scoring churches feature three repeated undulating triangular contours, where the central tower's pointed apex creates a striking vertical dimension. Medium-scoring examples showcase elongated gable walls with the tower's height matching the gable ridges, blending horizontal expansion with vertical tower presence. Low-scoring churches exhibit a two-part composition of roof and body, with the rectangular structure emphasizing horizontal proportions.

**Characteristics of church facade style 1.** Statistical analysis of facade style 1 characteristics from 62 churches in the Sichuan-Chongqing region reveals that scores are predominantly concentrated in two ranges: 6–8 points and 1–3 points, with an average of 5.74 points (Fig 4c). This indicates that most churches exhibit Western-style facade characteristics. Among them, there are 46 churches with a score of 5 or above, accounting for 74%, and their facade style is obviously inclined to Western style; while there are 16 churches with a score of less than 5, accounting for 26%, and their facade style is inclined to Chinese-style. This demonstrates that Western-style churches significantly outnumber Chinese-style ones in the Sichuan-Chongqing region.

Among these churches, the facade style scores range from 1 to 8.22 points, with a median of 6.54 and a minimum of 1.5. Comparative analysis reveals that the High-scoring churches feature gable-style facades adorned with arched windows and doors, along with Western-style column capitals, demonstrating a strong Western architectural influence. This trend occurs in regions where churches wield significant influence or where local resistance to religious practices is minimal [16]. Medium-scoring churches adopt Chinese-style archway designs with Western-inspired curved roof ridges, blending traditional Chinese elements like plaques and couplets with Western-style doors to create a hybrid aesthetic. Low-scoring churches closely resemble local residential buildings, featuring sloping roofs, blue-gray tiled roofs and Chinese-style rectangular wooden doors, all of which strongly emphasize their Chinese architectural identity.

**Characteristics of church facade style 2.** Statistical analysis of the evaluation scale scores for facade style 2 in 62 churches across the Sichuan-Chongqing region revealed that most scores fell within the 2–7 range (Fig 4d). Among them,

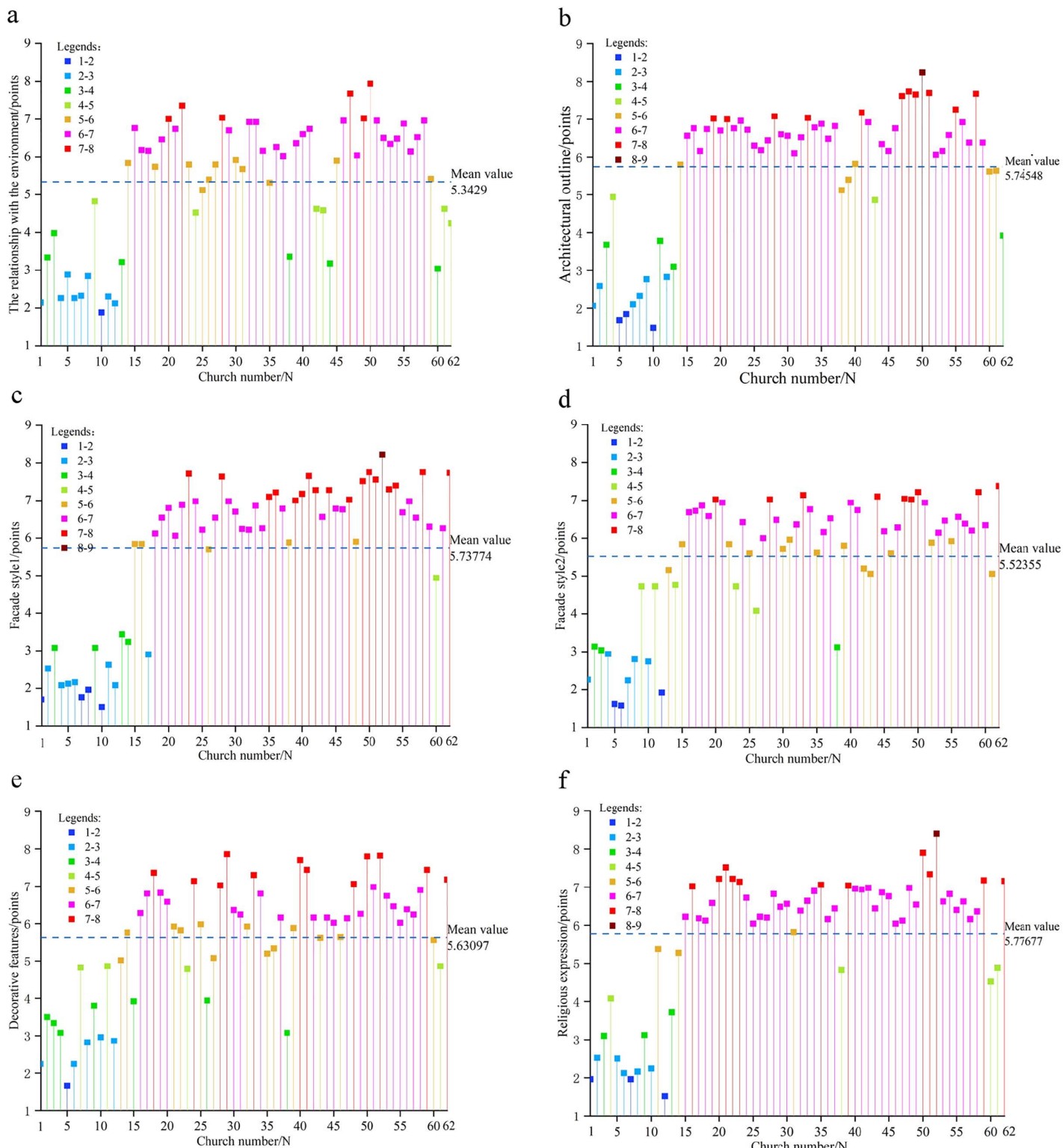

**Fig 4. Numerical analysis of evaluation factors for the church facade. (a)** Relationship between the church facade and environment; **(b)** Church facade contour; **(c)** Church facade style 1; **(d)** Church facade style 2; **(e)** Church facade decorative; **(f)** Church facade religious expression.

there are 46 churches with a score of 5 or above, accounting for 74%, and their facade style is inclined to official style; there are 16 churches with a score of less than 5, accounting for 26%, and their facade style is inclined to folk style. The average score for facade style 2 was 5.52, with significantly more churches displaying official architectural characteristics than those with folk influences.

Among these churches, the facade style 2 demonstrates a maximum score of 7.38, a median of 6, and a minimum of 1.58. Comparative analysis reveals distinct characteristics: High-scoring churches features a well-organized layout with clear hierarchy, vibrant wall colors, intricate door and window designs, exquisite decorations, and finely carved capitals. Medium-scoring churches showcases staggered columns and composite doorposts with railings, though its walls use simple paint and basic rose windows. Low-scoring churches employs local residential construction methods, including sloping roofs with small blue tiles, rammed earth walls, and wooden structures.

**Characteristics of church facade decoration.** Statistical analysis of facade decoration scores from 62 churches in the Sichuan-Chongqing region reveals two predominant score ranges: 5–8 points and 2–4 points (Fig 4e). Among them, there are 45 churches with facade decoration score above 5 points, accounting for 73%, and the decoration is relatively complex; there are 17 churches with facade decoration score below 5 points, accounting for 27%, and the decoration is relatively simple. The average decoration score was 5.63, demonstrating a significant disparity where more churches exhibit intricate decoration compared to those with simpler designs.

The facade decorations of churches scored a maximum of 7.86 points, a median of 6.02 points, and a minimum of 1.66 points. Comparative analysis reveals that the high-scoring churches extensively employ ceramic inlay decorations. These intricate designs not only adorn column capitals, lintels, and door arches but also ingeniously create vase motifs and natural landscape scenes on walls. Churches with median scores feature pointed column capitals, small spires, and bird motifs, complemented by text decorations on walls and lintels. The least decorated churches display only unremarkable wooden lintel decorations on their facades.

**Characteristics of church religious expression.** Facades Statistical analysis of the evaluation scale scores for religious expression characteristics in 62 churches across the Sichuan-Chongqing region reveals that most scores cluster in two ranges: 2–4 and 6–8 (Fig 4f). Among them, there are 47 churches with a facade religious expression score of 5 or above, accounting for 76%, and their religious doctrines are directly expressed. There are 15 churches with a facade religious expression score below 5, accounting for 24%, and their religious doctrines are implicit. The average score of 5.78 for religious expression indicates that directly articulated churches significantly outnumber those with implicit doctrinal expression in the Sichuan-Chongqing region.

Among these churches, the maximum score for facade religious expression is 8.4 points, the median is 6.4 points, and the minimum is 1.52 points. It can be concluded that the church with the high-scoring adopts the style of a Western-style church, featuring intricate carved window lattices, door lintels adorned with sheep and grass motifs expressing religious doctrines, and walls inscribed with the English letters "VENITE AD ME OMNES," which strongly conveys the direct expression of religious teachings. The churches with the median score employs the vertical three-part composition of a Western-style church, but its side towers adopt the form of a Chinese gable roof, with the facade engraved with the four characters "Jesus Christ" and equipped with Western-style elements such as round windows and shutters, resulting in a religious expression that lies between directness and subtlety. The church with the lowest score uses the gable entrance as the main facade entrance. In China, gable walls of residential buildings are typically side facades, while Western churches have a main entrance on the west facade, which serves as the main facade. This imitation of the main side facade entrance in the West subtly and uniquely expresses religious doctrines.

## Analysis of average comprehensive scores for Catholic church facades in Sichuan-Chongqing region

Statistical analysis of 62 churches in the Sichuan-Chongqing region across six evaluation factors reveals significant score variations among these institutions (Fig 5). This section calculates the sum and average scores of each evaluation factor

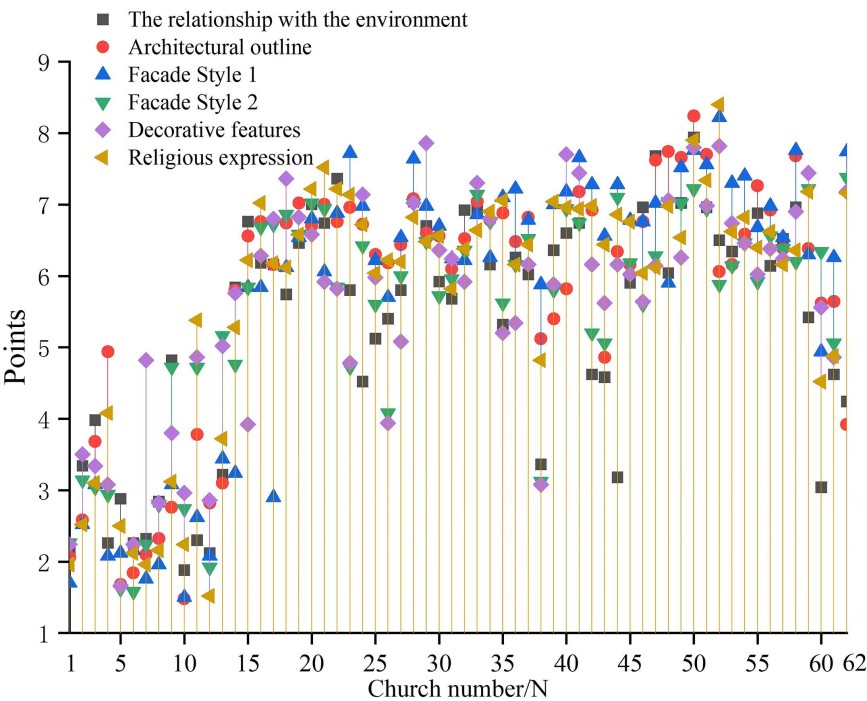

**Fig 5. Each score for 62 churches.**

for the 62 churches to determine their overall orientation toward Chinese or Western architectural styles. Churches are further categorized by their average scores, with an in-depth exploration of localization characteristics and integration approaches between Chinese and Western elements. The results show that the comprehensive average score of the 62 churches is 5.62624, indicating a general preference for Western-style characteristics. Specifically, 14 churches scored between 2–5, account for 22.6% showing predominantly Chinese features; 8 churches scored 5–6, account for 12.9%, exhibiting slightly Western-leaning characteristics; 34 churches scored 6–7, account for 54.8%, demonstrating clearly Westernized features; and 6 churches scored 7–8, account for 9.6%, demonstrating strong Western influences (Fig 6).

**Churches with comprehensive average scores of 2–5.** The average score ranges from 2 to 5, and the comprehensive Chinese-style churches include N1-N13, ranked as follows: N6 < N1 < N5 < N10 < N12 < N8 < N7 < N2 < N4 < N3 < N9 < N11 < N13 < N38 (Fig 8). From Fig 7a, it can be observed that churches scoring below 5 points in the comprehensive average demonstrate similar performance across all indicators. These churches exhibit integrated features with their surroundings: single-story facades and church heights nearly matching the surrounding trees. Their horizontal contours feature distinct two-part compositions of roof and body, presenting rectangular horizontal extensions. Chinese architectural elements are prominent, featuring walls constructed with traditional wooden boards and rammed earth materials, and facades showcasing Chinese wooden structures such as wooden columns, arch braces, and hanging flower columns. Many also adopt folk-style designs resembling local dwellings, featuring simple white, yellow, and wood-colored walls paired with gray-tiled gable roofs. Decorations are relatively minimal, primarily consisting of arch braces under eaves, hanging flower columns, lintel decorations, and geometric window lattice patterns. Religious symbolism is subtly expressed through limited motifs, often carved discreetly in window frames, lintels, arch braces, and other elements.

However, the curves of N38, N11, N13, and N9 exhibit notable anomalies. This analysis reveals that Chinese-style architectural integration in churches with facade composite scores between 2 and 5 can be categorized into four

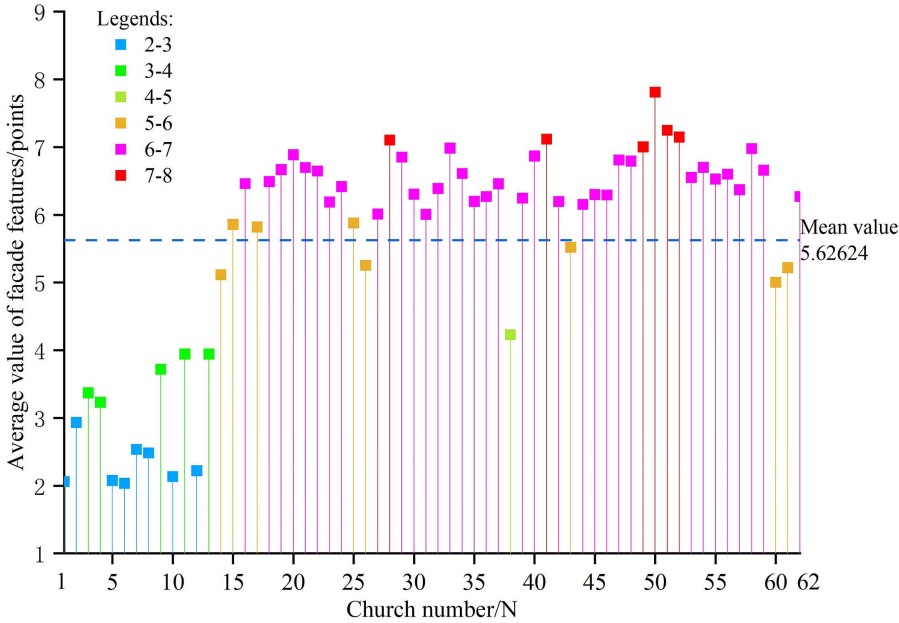

**Fig 6. Average scores of 62 churches.**

approaches: (1) Incorporating Western-style gable walls with arched windows and doors. (2) Employing polygonal column bases adorned with red, white, and black decorations featuring animal carvings, complemented by exquisite rose windows composed of 16 pointed arches and diamond-shaped lattice patterns; the "Cathedral of the Virgin Mary" plaque, rose windows, and crosses directly convey religious symbolism. (3) Utilizing gable walls as primary facades with sunflower motifs, taotie decorations, and intricate window patterns like Hui character designs, while incorporating Western elements such as crosses and arched windows to express religious teachings. (4) Employing asymmetrical horizontal compositions with vibrant wall coatings and Western-style pointed arch windows that directly communicate doctrinal messages.

**Churches with comprehensive average scores of 5–6.** The churches with comprehensive average scores of 5–6 include N60, N14, N61, N26, N43, N17, N15, and N25. These churches exhibit a general Western architectural inclination, with their score rankings as follows: N60 < N14 < N61 < N26 < N43 < N17 < N15 < N25 (Fig 9). While these churches share similar proportions and religious expressions, they demonstrate significant differences in environmental integration, architectural style, and decorative elements, which preliminarily reflect the selective absorption and innovative fusion of Chinese and Western architectural cultures. These churches feature two or three-story heights, with most exceeding surrounding vegetation to emphasize environmental prominence. Their silhouettes primarily consist of gable walls or cascading eaves, presenting a vertically aligned profile. They incorporate both Chinese archway and Western gable architectural elements, generally adhering to official architectural conventions with distinct primary and secondary facades. Decorative features include plaque inscriptions, couplets, floral patterns, rose windows, and false windows. Religious expressions are conveyed through both the direct use of Western-style church exteriors and decorative motifs, as well as subtle textual interpretations in couplets and plaques.

As shown in Fig 7b, the six-point evaluation criteria for these churches reveal outliers in the N14, N17, N60, N15, and N26 curves. This indicates that the overall characteristics of this average score range reflect a fusion of Chinese and Western architectural styles in Western-style churches, characterized by: (1) Chinese-style facade design featuring archway elements; (2) Integration with the environment through surrounding tall trees; (3) Minimalist decoration with only cross

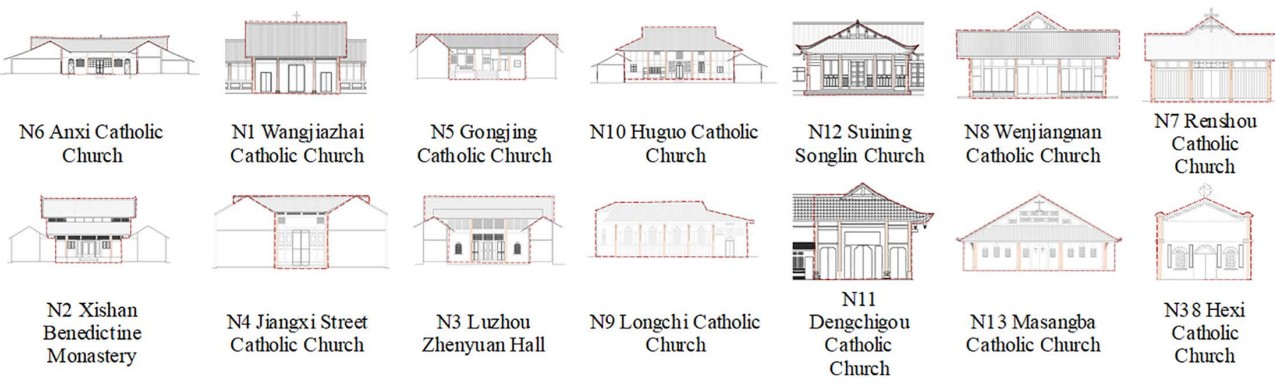

**Fig 7. Church facade scoring based on average segment analysis. (a)** Comprehensive average scores of 2-5; **(b)** Comprehensive average scores of 5-6; **(c)** Comprehensive average scores of 6-7; **(d)** Comprehensive average scores of 7-8.

N6 Anxi Catholic Church

N1 Wangjiazhai Catholic Church

N5 Gongjing Catholic Church

N10 Huguo Catholic Church

N12 Suining Songlin Church

N8 Wenjiangnan Catholic Church

N7 Renshou Catholic Church

N2 Xishan Benedictine Monastery

N4 Jiangxi Street Catholic Church

N3 Luzhou Zhenyuan Hall

N9 Longchi Catholic Church

N11 Dengchigou Catholic Church

N13 Masangba Catholic Church

N38 Hexi Catholic Church

**Fig 8. Churches with average scores between 2 and 5.**

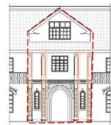
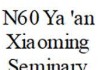
N60 Ya'an Xiaoming Seminary

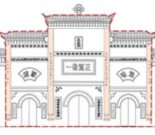
N14 Seven Duiwa Catholic Church

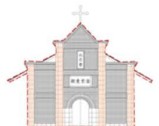
N61 Benevolent Catholic Church

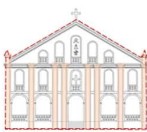
N26 West Street Catholic Church

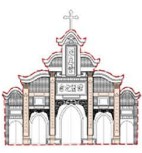
N43 Yongchuan Saint Michael Church

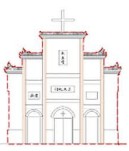
N17 Zhengdongjie Catholic Church

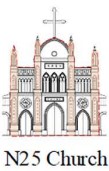
N15 Mugu Catholic Church

N25 Church of Our Lady of the Rosary

**Fig 9. Churches with average scores between 5 and 6.**

motifs on the ridge finial, lacking other religious symbolism; (4) Strong folk architectural influence demonstrated by rectangular tile cladding and plaque decorations.

**Churches with comprehensive average scores of 6–7.** There are 34 churches with an average composite score of 6–7, including N16, N18-N24, N27, N29-N37, N39, N40, N42, N44-N48, N53-N59, and N62, which constitute the largest group in terms of quantity. These churches generally exhibit Western architectural characteristics with minimal score variations, as shown in the ranking illustrated in Fig 10. They demonstrate distinct Western features across multiple aspects: most buildings are two or three stories tall, typically exceeding surrounding vegetation. Their architectural contours predominantly feature vertical compositions with diverse styles including tiered archways, Western-style gables, staggered gable designs, parallel or sloping side walls, and tower-like structures that create upward verticality. The architectural styles lean towards Western and official styles, with most churches displaying pronounced Western influences, such as vertical three-part wall divisions through columns. Official style characteristics are prominent, showcasing exquisite craftsmanship. Decorative elements are rich and varied, encompassing religious symbols like lettering, sculptures, animal motifs, stained glass, crosses, three semicircular symbols, and scrollwork. Religious doctrines are expressed directly through elaborate decorative patterns, intricate carvings, architectural designs, inscriptions, and resplendent color schemes that clearly convey spiritual messages.

As shown in Fig 7c, the SD curve scores of each church reveal outliers at N44, N62, N23, N24, and N42. This analysis reveals the distinctive characteristics of Western-style churches with Chinese influences in this average score range: (1) The chapel occupies the central position of the architectural complex, featuring a two-story structure with a monastic layout. It is surrounded by trees of equal height, creating a harmonious integration with the environment; (2) The facade adopts a vertical five-section design, extending horizontally on both sides of the Western-style three-section composition, with a horizontal contour; (3) The vernacular style is characterized by white-painted walls with uniform door and window designs, minimal ornamentation, and only decorative window lattices and carved motifs of sheep and grass on door lintels, which blend seamlessly with the white background; (4) The gable ends are flat rather than pointed, with horizontal contours matching the surrounding vegetation height for better environmental integration; (5) The compact scale and surrounding vegetation create a harmonious blend with the surroundings.

**Churches with comprehensive average score of 7–8.** The churches with composite average scores of 7–8 include N28, N41, N49, N50, N51, and N52 (Fig 11). These churches exhibit prominent environmental characteristics in two aspects: Firstly, they are significantly taller than their surroundings; secondly, their architectural styles differ markedly from the local environment. Their silhouettes possess a distinct verticality, reinforced by either the overall or partial mountain-like contours and the towering spires of the churches themselves. The Western architectural style is evident in their facade designs, which closely resemble those of Western churches. The official architectural style is clearly manifested through exquisite craftsmanship and the use of premium materials of the era. The decorations are exceptionally intricate, featuring not only traditional Chinese inscriptions and couplets on the facades but also Western elements such as

N16 Bishan Lude Hall

N18 Xuan Yi Rose Academy

N19 Yuantong Catholic Church

N20 Xichang Catholic Church

N21 Anfu Catholic Church

N22 Jiangzhou Catholic Church

N23 Xiushui Catholic Church

N24 Xiaojin Catholic Church

N27 Miaoyu Catholic Church

N29 Gongxing Street Catholic Church

N30 Zizhong Catholic Church

N31 Qionglai Wushengtang

N32 Yongjia Catholic Church

N33 Bachuan Catholic Church

N34 Dechang Sacred Heart Church

N35 Shaba Catholic Church

N36 Shifang Shengxiu Church

N37 Huili Catholic Church

N39 Zhangjiaxiang Catholic Church

N40 Shujiawan Catholic Church

N42 Zhenyuan Hall, Shima Township

N44 Cimushan Monastery

N45 Fuling Catholic Church

N46 Yangliu Street Catholic Church

N47 Delesa Catholic Church

N48 Moxi Catholic Church

N53 Mianzhu Catholic Church

N54 Ya'an Church of Our Lady

N55 Hebaochang Catholic Church

N56 Nanchong Catholic Church

N57 Wenxingjie Catholic Church

N58 lingbao Seminary

N59 Tongguanyi Catholic Church

N62 Ping'an qiao Catholic Church

**Fig 10. Churches with average scores between 6 and 7.**

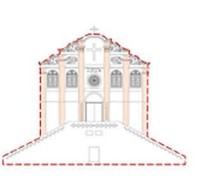

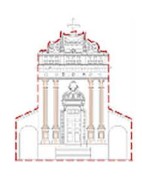

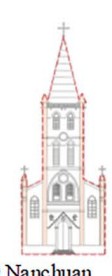

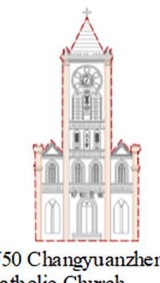

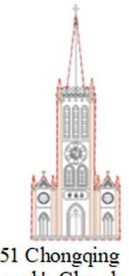

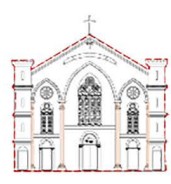

N28 Helong Catholic Church

N41 Zhongxian Tianchi Monastery

N49 Nanchuan Catholic Church

N50 Changyuanzhen Catholic Church

N51 Chongqing Joseph's Church

N52 Berlin Catholic Church

**Fig 11. Churches with average scores between 7 and 8.**

scrollwork window patterns, false windows, bells, Corinthian capitals, and vase motifs. Through forms nearly identical to Western churches, these structures directly convey religious doctrines.

These churches demonstrate balanced scores across six evaluation criteria (Fig 7d). Characterized by Western-style pews integrated with Chinese architectural elements, they feature gable-and-twin-tower facades where the towers' lower height creates a subtle visual contrast with the surrounding environment. The horizontal placement of twin towers further diminishes the vertical prominence of their silhouette. Additionally, the uniform white-painted walls partially obscure the distinctive feature of the official-style architecture.

### Correlation analysis of evaluation factors for facade

The aforementioned research demonstrates significant correlations among evaluation factors, with only a few showing outliers in correlation analysis. Building on this foundation, this section first conducted a comprehensive correlation analysis on all numerical values from the collected evaluation scales to thoroughly examine overall inter-factor relationships. Subsequently, correlation analysis was performed on the calculated means of each church factor to meticulously examine their interconnections, thereby providing a more precise theoretical basis for analyzing the structural characteristics of church facades.

**Correlation analysis based on all numerical data.** This analysis utilized a rating scale for 62 churches, collecting 50 valid samples that yielded 3,100 data points. Prior to the correlation analysis, a Kolmogorov-Smirnov (K-S) test was conducted to determine whether the data followed a normal distribution, which is a prerequisite for selecting the appropriate statistical method. The K-S test results showed all variables had K-S statistics between 0.188 and 0.236 (Fig 12), With p-values all below 0.05. indicating significant deviations from normal distribution. This indicates a significant departure from a normal distribution, suggesting that the data does not meet the assumptions required for parametric testing. Given these results, Spearman's correlation coefficient was selected as it is a robust non-parametric measure specifically designed for non-normally distributed or ordered categorical data. Using SPSS software, we tested the correlation between the six factors. The results show that in the comprehensive evaluation of the main facade characteristics of churches in Sichuan and Chongqing region, the correlation coefficient ρ between each factor is between

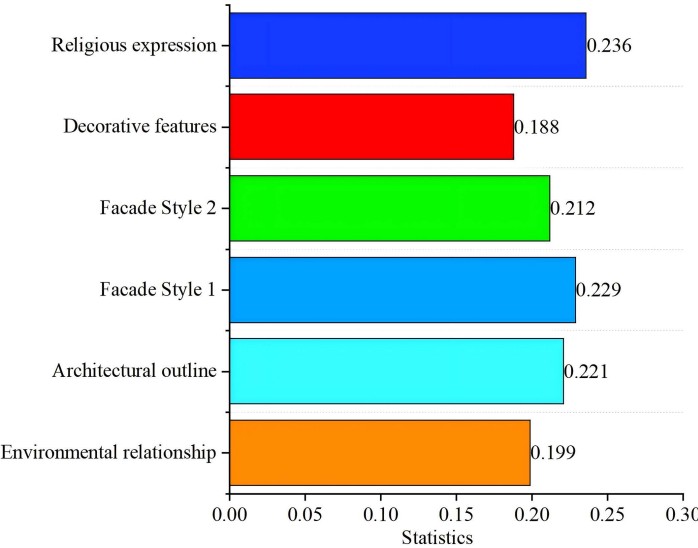

**Fig 12. Statistical quantity Dn from the K-S normality test for all numerical data.**

0.582 and 0.728 (Fig 13). The correlation coefficient between environmental relationship, architectural outline, facade style, decorative characteristics and religious expression is all greater than 0.5, showing positive correlation.

The strongest correlations were observed between:

The most strongly correlated factors are as follows ($\rho > 0.7$): facade style 1 demonstrates a 0.713 correlation with religious expression, while facade style 2 exhibits a 0.728 correlation with decorative features. These two factor groups demonstrate the strongest correlations. This indicates that Western-style churches tend to express religious doctrines more directly, whereas Chinese-style churches prefer subtle expressions. Official-style churches favor intricate decorations, while folk-style churches lean toward simple ornamentation.

Strongly correlated factors ($0.65 < \rho < 0.7$): The correlation coefficient between environmental relationships and architectural contours is 0.687, showing a strong correlation. This indicates that churches integrated with the environment mostly have horizontal architectural contours, while those protruding from the environment mostly present vertical architectural contours.

Factors with relatively weak correlations ($\rho < 0.6$): The correlation coefficient between environmental relationships and decorative features is 0.582, while that between environmental relationships and religious expression is 0.588. The correlation coefficient between decorative features and architectural contours is 0.591. These factors demonstrate relatively weak interconnections. This indicates that the integration of the church with its environment shows limited correlation with the complexity of decorations or the directness of religious expression. Moreover, the complexity of decorative elements does not determine whether the church's contours are horizontal or vertical.

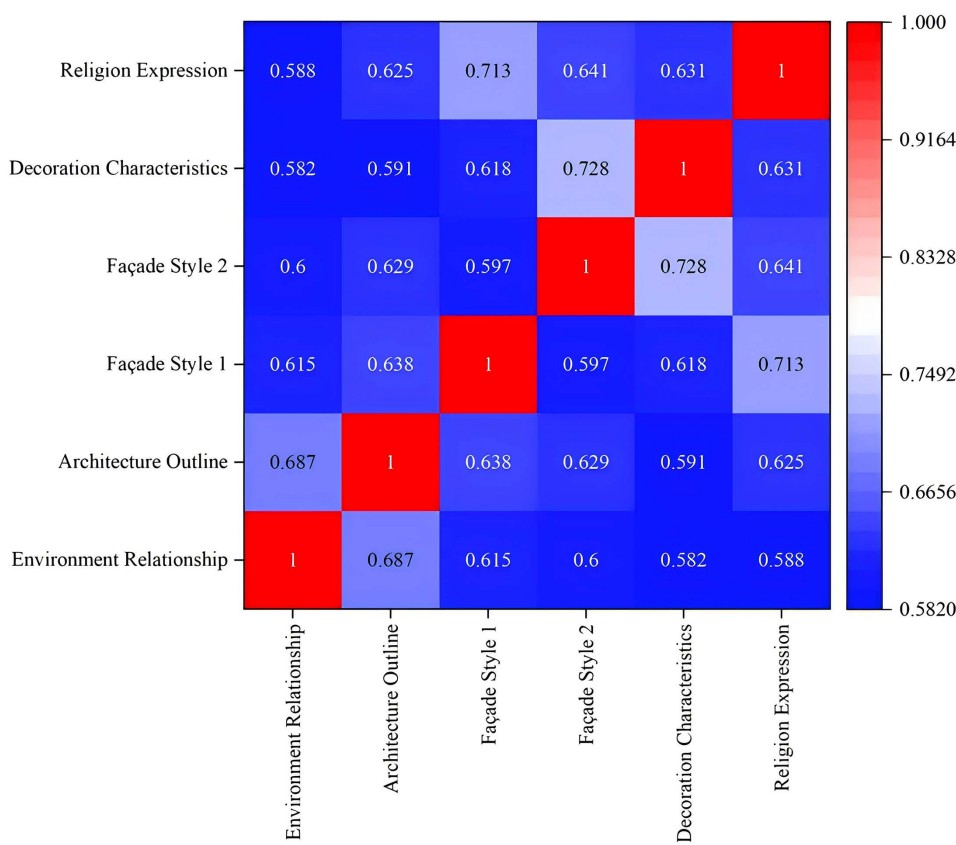

**Fig 13. Spearman's correlation coefficient results based on all numerical values.**

To sum up, the main facade of churches in the Sichuan and Chongqing region is closely related to the environment, contour, style, decoration, and religious expression, yet it also maintains a certain independence in a specific dimension.

**Correlation analysis based on calculated means.** To eliminate the discreteness of individual evaluations and further explore stable correlation patterns among the facade characteristics of churches, this study conducted correlation analysis using the calculated means of each church's factors. Each church served as a sample, with each sample containing six factor means, resulting in a total of 62 churches and 62 data records.

First, the K-S method was employed to test data normality (S1 Appendix Formula 3–6). The test results showed that all variables K-S statistics ranged between 0.146 and 0.270 (Fig 14). With p-values consistently below 0.05, the results indicated a significant departure from a normal distribution, suggesting that the data did not meet the prerequisites for parametric testing. Given that the data were ordinal and non-normally distributed, this section utilized Spearman's correlation analysis in SPSS software to examine the correlations among the six evaluation criteria. Calculation results revealed that all factor correlation coefficients ($\rho$) ranged between 0.597 and 0.857 (Fig 15), with p-values all below 0.05. This demonstrates that in evaluating the main facade styles of churches in the Sichuan-Chongqing region, factors such as environmental relationship, facade contour, architectural style, decorative elements, and religious expression all exhibited positive correlations, with coefficients exceeding 0.55.

Compared with the correlation analysis based on all numerical values, the overall positive correlation relationship remains unchanged, but the correlation coefficients have shown an overall improvement, indicating that the overall synergistic trend between factors becomes more stable after eliminating individual differences. Specifically, the correlations between factors are manifested as follows: The three groups with stronger correlations further enhanced their coefficient values, but the group with the strongest correlation changed. The correlation coefficient between environmental relationships and architectural contours increased from 0.687 to 0.776, while the correlation coefficient between facade style 1 and religious expression rose from 0.713 to 0.754, both demonstrating strong correlations. The correlation coefficient between facade style 2 and decorative features increased from 0.728 to 0.857, which is considered the strongest correlation combination. The mean-based analysis effectively reduces the dispersion of individual scores, thereby more clearly

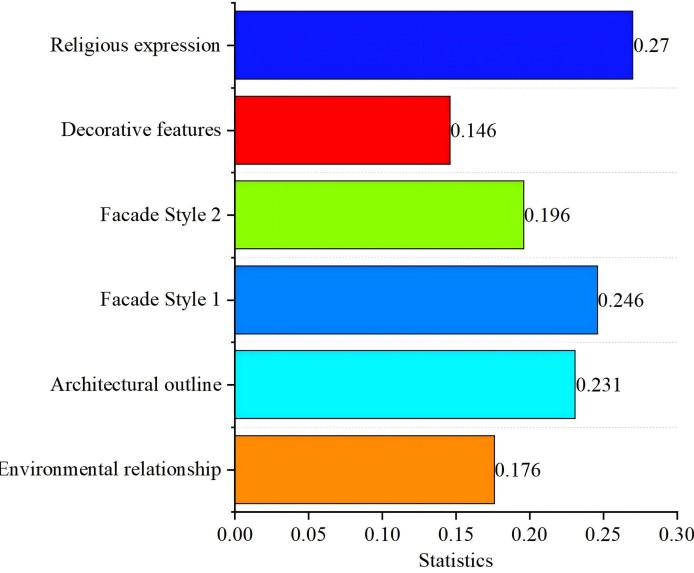

**Fig 14. Statistical quantity Dn from the K-S normality test based on mean calculation.**

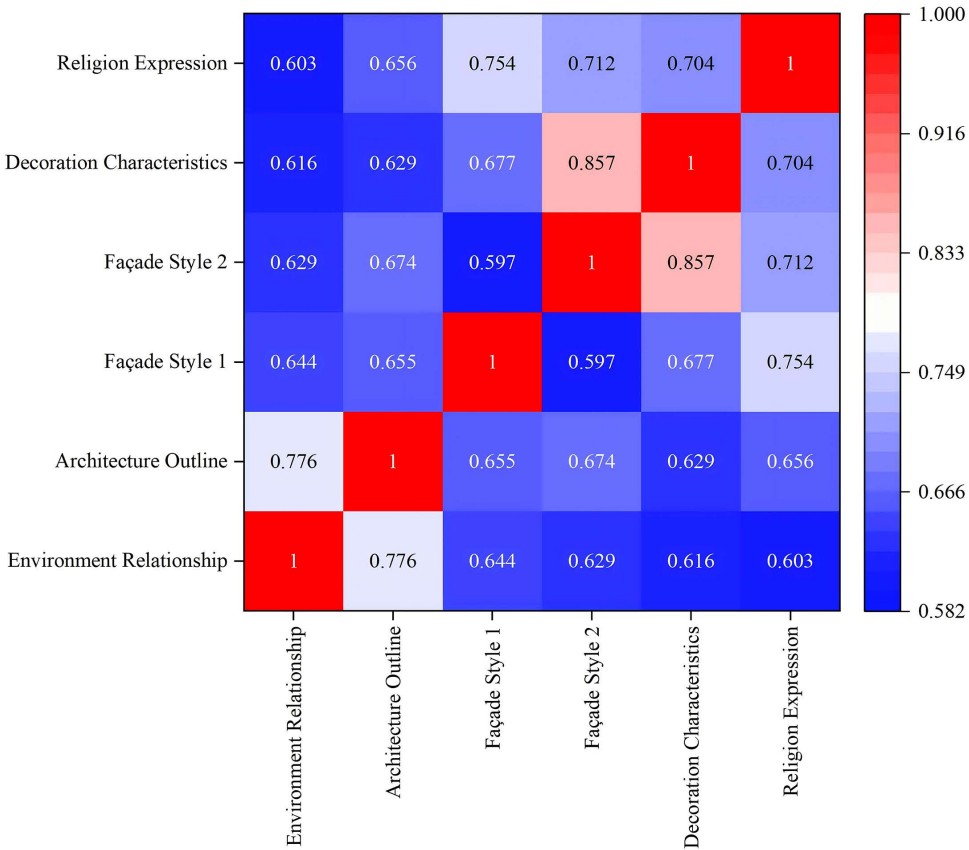

**Fig 15. Spearman correlation coefficient results based on mean calculation.**

reflecting the overall associations between different factors. Particularly, the positive relationship between the tendency for folk or official style in facade design and decorative complexity becomes more prominent.

The three groups with relatively weak correlations showed increased correlation coefficients, though the weakest correlation group underwent a change. The correlation coefficient between facade-environment relationship and decorative features is 0.616; the correlation coefficient between facade-environment relationship and religious expression is 0.603; and the correlation coefficient between decorative features and architectural outline is 0.629. All three groups show relatively weak correlation. Meanwhile, the correlation coefficient between facade style 1 and style 2 remained unchanged at 0.597, becoming the group with the weakest correlation. This indicates that decorative features and religious symbolism are more closely associated with internal design characteristics rather than external environmental adaptability, a pattern that remains consistent even in the mean value analysis. Additionally, the correlation between facade style 1 and style 2 was already weak at the individual evaluation level, with the correlation coefficient remaining unchanged after mean value analysis. This indicates that these two stylistic dimensions within the facade system represent relatively independent evolutionary paths or distinct aesthetic categories, rather than being governed by a singular consistent logic.

In conclusion, the correlation analysis based on mean calculation further verified the significant positive correlation between the factors of the main facade of modern churches in Sichuan and Chongqing. It also revealed the synergistic pattern and the relative independence among these factors more clearly by weakening individual differences.

## Discussion

Overall, the comprehensive characteristics of church facades in this study align with those of numerous scholars. Between 1840 and 1949, Western-style church models were constructed in China, while churches also underwent a process of localization. Influences on the Western characteristics of churches include, for example, "from 1927, the architectural works of Benedictine friar Adelbert Grisnieth became the official model for churches" [53]; the typical elevation of church buildings above surrounding structures, reflecting their central role in people's lives [54]; and the independent construction of Catholic churches in areas far from residential zones to create a sense of sanctity [9]. Influences during the localization process include, for instance, Archbishop Jean-Baptiste Regis, who served as the Apostolic Representative in China from 1922 to 1933, publicly criticizing Gothic and Roman styles and advocating for a "Chinese-Western fusion" in art and architecture, leading to a gradual weakening of Western architectural uniformity [55]. Moreover, pioneering missionaries actively adapted to Chinese culture. Most churches built in China before 1900, except those in foreign concessions, were modest in scale and used local building materials and techniques. These churches were the result of collaboration between missionaries and Chinese believers, constructed according to principles of cultural adaptation and exchange, reflecting a clear tendency toward architectural localization and cultural adaptation during this period [56,57]. Moreover, in actual construction, the Western church prototype has been continually reinterpreted by local craftsmen [4].

This study conducted a comprehensive analysis of the facade characteristics of 62 Catholic churches in the Sichuan-Chongqing region using the semantic difference method. The research revealed that these churches, as foreign cultural structures, have achieved architectural localization in five aspects: the relationship between the church and its environment, architectural silhouette, architectural style, decorative elements, and religious expression. Regarding the relationship between churches and their surroundings, the church's height harmonizes with surrounding residential buildings and trees, while the bell tower's height aligns with the main structure, creating seamless integration. The architectural silhouette adopts horizontal proportions, featuring elongated mountain-shaped contours and rectangular cross-sections. In terms of style, the churches utilize traditional Chinese archways and residential-style architecture, employing understated colors and locally sourced materials without hierarchical treatment of facades. Decorative elements are kept minimal and discreetly positioned. For religious expression, the church facades primarily feature traditional Chinese gable walls, subtly conveying religious teachings through architectural design.

For future church architecture blending Chinese and Western elements, the following integrated design strategies are proposed. In Chinese-style churches, incorporate Western features like arched windows, rose windows, and plaques displaying church names. Adorn walls with couplets praising Catholic teachings, cross-shaped fish motifs, and vibrant cross decorations. Integrate cross motifs into inconspicuous areas, including window frames, lintels, and arches. The main facade features a polygonal altar area on the left side, with the entrance positioned at the right end. For Western-style churches, ensure harmonious integration with surroundings through delicate facades and lower-tower designs. Emulate the layered descending patterns of traditional Chinese archways. Extend the architectural silhouette horizontally using a Western three-part composition, with additional facade extensions on both sides. Maintain minimalist decoration using only color accents and rectangular ceramic tiles, with religious teachings conveyed solely through inscribed plaques. The comprehensive evaluation of the characteristics of church facades in Sichuan and Chongqing helps the administrative departments to implement the policy of religious Sinicization more effectively, and promote the further exploration of the unique Chinese and Western fusion culture in the cultural heritage.

In practical implementation, the following Sino-Western integration strategy is proposed. Under the Sinicization policy of Christianity, as religious expression carries the highest score among six evaluation factors, future cultural management departments should focus more on how to localize the religious doctrine expression in church facades. Since religious expression is closely tied to the Chinese or Western architectural style of churches, efforts could begin with the localization of facade styles. Additionally, architectural heritage departments should delve into the implicit emotional expressions embedded in church facade characteristics when examining their features. Regarding community-level church utilization,

emphasis should be placed on incorporating localized and secular elements to better integrate churches as public facilities into communities. On the identity influence level, for buildings occupied by Western residents, Western elements can be integrated while maintaining overall Chinese architectural style to strengthen their sense of identity.

This study has the following limitations: First, the number of participants invited to complete the evaluation scale was limited. Second, all survey participants were architecture students, whose perspectives may differ from those of experts or the general public. Therefore, future research should leverage the internet to recruit more participants, making the data analysis results more representative. Additionally, it would be beneficial to include diverse groups such as religious believers, non-believers, experts, scholars, and ordinary citizens in evaluating church facades.

## Conclusion

This study employs the semantic difference method for the comprehensive feature evaluation of church facades, offering a novel perspective and effective approach for the research on architectural heritage facades. By systematically analyzing facade characteristics across five dimensions—comprised of six specific criteria: environmental relationship, facade contour, facade style 1 (Chinese vs. Western), facade style 2 (Folk vs. Official), facade decoration, and religious expression, the theoretical scope of the research on the church facade system has been further broadened.

From a regional perspective in Sichuan-Chongqing, 22.6% of churches exhibit predominantly Chinese architectural characteristics. Among the six evaluation criteria, the facade's religious expression demonstrates the lowest degree of Chinese influence. However, it also achieves the highest score in religious expression among all factors. These findings provide valuable insights for implementing policies on religious Sinicization, highlighting how the visual presentation of religious doctrines on facades serves as a significant indicator of architectural Sinicization.

From the perspective of individual church buildings, the integration of Chinese and Western cultures in the facades of churches in the Sichuan-Chongqing region is diverse. Churches with an overall Chinese style tend to incorporate elements such as Western-style arched doors and windows, cross patterns or components, plaques and couplets inscribed with Western religious doctrines, and polygonal contours. In contrast, churches with an overall Western style adopt Chinese architectural and environmental integration concepts, reducing the volume and height of the church, adopting a Chinese archway style, enhancing the horizontal contour of the building, and using China's building materials. This conclusion can serve as a reference for cultural management departments to explore unique Chinese and Western cultural integration. The correlation analysis of facade factors based on all numerical and mean values reveals that whether a church's facade adopts an official or folk style is closely related to the complexity of its decorations. Similarly, the integration of the church with its environment is closely linked to whether its silhouette exhibits horizontal or vertical characteristics. This reveals consistent patterns of association between facade features, suggesting that certain stylistic elements tend to co-occur in the architectural composition of these churches.

In summary, in the work of architectural heritage preservation, we should not only pay attention to the explicit cultural characteristics of its facade, but also emphasize the implicit emotional expression features of its facade. In addition, as a carrier of foreign culture, the architectural language of these churches demonstrates a sophisticated integration of localized features. A profound understanding of their comprehensive facade characteristics is essential for preserving the unique identity of this heritage within its regional context. In future research, given the limitations of this paper, greater attention should be paid to the cognition of church facades by different groups of people. Based on the findings of this study, it is hoped that more localized expressions of religious doctrines in church facade characteristics can be explored in the future, promoting the exchange and integration of Chinese and Western cultures.

## Supporting information

**S1 Appendix. Calculation formulas.**
(DOCX)

**S1 Tabel. Aggregated mean scores of the six evaluation criteria for 62 churches.**
(XLSX)

## Acknowledgments

I sincerely thank my senior colleagues in the research group for their participation in the church field research, which enabled me to obtain substantial firsthand materials on churches.

## Author contributions

**Conceptualization:** Qin Lai, Yunzhang Li.

**Data curation:** Xinkai Li.

**Formal analysis:** Beibei Zhang.

**Investigation:** Beibei Zhang.

**Methodology:** Yunzhang Li.

**Software:** Qin Lai, Nan Chen.

**Supervision:** Xinkai Li, Nan Chen, Yunzhang Li.

**Validation:** Qin Lai.

**Writing – original draft:** Qin Lai.

**Writing – review & editing:** Yunzhang Li.

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
