## [Decision Letter · Decision Letter 0]

15 Feb 2026

PONE-D-25-67492Comprehensive feature evaluation of the main facades of Catholic churches in Sichuan-Chongqing region based on semantic difference method, 1840-1949PLOS One

Dear Dr. Li,

Thank you for submitting your manuscript to PLOS ONE. After careful consideration, we feel that it has merit but does not fully meet PLOS ONE’s publication criteria as it currently stands. Therefore, we invite you to submit a revised version of the manuscript that addresses the points raised during the review process.

Please check the comments from the two reviewers and pay attention to the critical issues they have raised.

We look forward to receiving your revised manuscript.

Kind regards,

Kofi Agyekum, Ph.D.

Academic Editor

PLOS One

Journal Requirements:

Additional Editor Comments:

Dear Author,

Thank you for considering PLOS-ONE for your manuscript. After undergoing reviews, the reviewers agree that the manuscript is well-improved. However, there are some minor issues that must be resolved. Please check the issues they have raised and revise your manuscript as such. I wish you luck with this revision.

Reviewers' comments:

Reviewer's Responses to Questions

**Comments to the Author**

1. Is the manuscript technically sound, and do the data support the conclusions?

Reviewer #1: Partly

Reviewer #2: Yes

2. Has the statistical analysis been performed appropriately and rigorously?

Reviewer #1: Yes

Reviewer #2: Yes

3. Have the authors made all data underlying the findings in their manuscript fully available?

Reviewer #1: Yes

Reviewer #2: Yes

4. Is the manuscript presented in an intelligible fashion and written in standard English?

Reviewer #1: Yes

Reviewer #2: Yes

5. Review Comments to the Author

Reviewer #1: The manuscript is partially sound; conclusions are generally supported but occasionally over-extended. It presents a technically coherent study that applies the semantic difference (SD) method in a systematic and replicable manner. The research design is clearly described, the evaluation dimensions are grounded in prior literature, and the sample size of 62 churches × 50 evaluators (3,100 data points) is adequate for perceptual analysis of this type. Reliability and validity checks (Cronbach’s Alpha, KMO, Bartlett’s test) indicate acceptable to strong internal consistency.

The descriptive results and correlation analyses broadly support the authors’ main conclusions regarding:

i. The predominance of Western-oriented facade characteristics.

ii. The existence of localized Sino–Western hybridization.

iii. Positive associations between facade style, decoration, and religious expression.

However, some interpretive conclusions particularly those linking facade characteristics to missionary intent, social resistance, or cultural strategy go beyond what the perceptual data alone can substantiate. These claims would require stronger historical or archival triangulation. With more cautious phrasing, the conclusions would be fully aligned with the presented data.

The statistical analysis is appropriate for the nature of the data and is generally rigorous:

i. The use of Cronbach’s Alpha, KMO, and Bartlett’s test appropriately establishes scale reliability and suitability for multivariate analysis.

ii. Normality testing via the Kolmogorov–Smirnov test is correctly applied, and the subsequent choice of Spearman’s rank correlation for non-normal, ordinal data is methodologically sound.

iii. The two-tier correlation analysis (all data points vs. church-level means) is a strength, as it helps reduce individual-level noise and confirms stability of relationships.

That said, the manuscript occasionally uses strong structural or causal language (“structural relationship,” “decision-making logic”) while relying solely on bivariate correlations. Correlation analysis alone cannot fully support structural claims. This does not invalidate the analysis but suggests the need for either: more cautious interpretation, or complementary analyses (e.g., factor analysis or clustering), if such claims are to be retained.

The manuscript provides: Summary statistics, Reliability and validity outputs, Correlation matrices, Graphical representations of distributions. But it is not fully clear whether the raw SD scores (i.e., individual participant ratings for each church and each adjective pair) are provided in a reusable format (e.g., spreadsheet or table in Supporting Information). While not strictly required, best practice under PLOS policy encourages availability of data points underlying means and correlations.

The authors should explicitly confirm that raw or semi-raw scoring matrices are included as Supporting Information, or provide them in a tabular format to enhance transparency, replicability, and secondary analysis.

Reviewer #2: The study addresses a topic that has historical, architectural, and religious significance. It is also relevant to the construction industry, particularly architecture, and has practical implications for cultural management departments and architectural heritage preservation. Additionally, the utilization of the semantic difference method to carry out the feature evaluation of the facades of the churches under study brings a novel dimension.

The manuscript exhibits several promising features, including its focus on the subject being investigated. The study contributes meaningfully to knowledge by addressing a methodological gap in earlier studies on comprehensive evaluations of facades of Catholic churches in the study area. Overall, the literature review presented in the literature review section is relevant and current to some extent. However, the following areas of the manuscript require attention to enhance its robustness.

1. The introduction section of the study, though concise, presents an adequate background to the study. It could have benefited from more recent citations. Also, there is no reference to literature between lines 70 and 75. Kindly rephrase the sentence between lines 83 and 85: "Furthermore, the relationship between facade structural characteristics is revealed from both the overall evaluation score of the church and the evaluation score of each individual church." The use of "the church" and "individual church" might not be clear to some audiences.

2. The sub-section, 3.1 (Research area) under the Research methodology section, provides the requisite information situate the study. However, the source of the literature and supporting Figure have not been provided. Also, the limitations regarding the study participants are clearly stated in the discussion section, but the sampling/selection method has not been described in the methodology.

3. The study employs the semantic difference method to evaluate the main facade characteristics of these churches across five dimensions, namely: relationship with the environment, facade contour, facade style, facade decoration, and religious expression. It has six evaluation criteria, subdivided into facade style 1 and façade style 2. This should be explicitly stated for clarity.

4. Under subsection 4.4. (Correlation analysis of evaluation factors for façade) in the analysis section, kindly rephrase the sentence between lines 550 and 553 “Subsequently, correlation analysis will be performed on the calculated means of each church factor……” The portion “will be performed” should be “was performed”.

Also, could lines 558 and 597-598, which present the data’s “violation and significant departure from normalcy be explained further for clarity?

5. Please, rectify the following references: 786 Coomans, T., & Luo, W. (2015); 789 Coomans, T. (2025) and 832 Ocker, C., & Elm, S. (2020).

Overall, the manuscript has many strengths, and the above suggestions, if considered/addressed, will further strengthen the paper.

6. PLOS authors have the option to publish the peer review history of their article (what does this mean?). If published, this will include your full peer review and any attached files.

Reviewer #1: **Yes:** Henry Kofi Dansu

Reviewer #2: No

---

## [Author Response · Author response to Decision Letter 1]

2 Apr 2026

Dear Reviewers,

We appreciate the opportunity to revise our manuscript titled "Comprehensive feature evaluation of the main facades of Catholic churches in Sichuan-Chongqing region based on semantic difference method, 1840-1949" and are grateful for the insightful comments provided by the reviewers.The comments are all valuable and very helpful for revising and improving our paper, as well as the important guiding significance to our researches. We have carefully addressed all the points raised by the reviewers. In the following section, we provide a point-by-point response to the comments. For clarity, the reviewers' comments are presented in italics, and our corresponding responses and revisions are highlighted in blue text. Additionally, all major changes in the revised manuscript are marked in red. We have conducted a comprehensive revision of the entire manuscript and hope that this version now meets the requirements for publication.

Response to Reviewer 1:

The manuscript is partially sound; conclusions are generally supported but occasionally over-extended. It presents a technically coherent study that applies the semantic difference (SD) method in a systematic and replicable manner. The research design is clearly described, the evaluation dimensions are grounded in prior literature, and the sample size of 62 churches × 50 evaluators (3,100 data points) is adequate for perceptual analysis of this type. Reliability and validity checks (Cronbach’s Alpha, KMO, Bartlett’s test) indicate acceptable to strong internal consistency.

The descriptive results and correlation analyses broadly support the authors’ main conclusions regarding:

i. The predominance of Western-oriented facade characteristics.

ii. The existence of localized Sino–Western hybridization.

iii. Positive associations between facade style, decoration, and religious expression.

(1)However, some interpretive conclusions particularly those linking facade characteristics to missionary intent, social resistance, or cultural strategy go beyond what the perceptual data alone can substantiate. These claims would require stronger historical or archival triangulation. With more cautious phrasing, the conclusions would be fully aligned with the presented data.

Response: Thank you for your valuable and constructive suggestions. We agree that our perceptual data, derived from the SD method, primarily reflects the visual and formal characteristics of the facades rather than the historical motivations or strategies behind them.

Following your suggestion, we have refined the Conclusion and Discussion sections using more cautious and descriptive phrasing. Specifically:We have shifted the interpretative focus from "missionary intent" and "cultural strategies" to "stylistic manifestations" and "architectural integration tendencies." We have removed/toned down claims regarding the socio-political motivations of the builders that were not directly supported by historical archives within this specific study. Please refer to the revised text in Lines 681-682, 745-747, and 765-768.

The statistical analysis is appropriate for the nature of the data and is generally rigorous:

i. The use of Cronbach’s Alpha, KMO, and Bartlett’s test appropriately establishes scale reliability and suitability for multivariate analysis.

ii. Normality testing via the Kolmogorov–Smirnov test is correctly applied, and the subsequent choice of Spearman’s rank correlation for non-normal, ordinal data is methodologically sound.

iii. The two-tier correlation analysis (all data points vs. church-level means) is a strength, as it helps reduce individual-level noise and confirms stability of relationships.

(2)That said, the manuscript occasionally uses strong structural or causal language (“structural relationship,” “decision-making logic”) while relying solely on bivariate correlations. Correlation analysis alone cannot fully support structural claims. This does not invalidate the analysis but suggests the need for either: more cautious interpretation, or complementary analyses (e.g., factor analysis or clustering), if such claims are to be retained.

Response: Thank you for your valuable and constructive suggestions. We agree that bivariate correlation analysis identifies associations rather than establishing definitive causal or structural relationships.

In response to your suggestion, we have opted for a more cautious interpretation of our findings throughout the manuscript. Specifically: We have replaced strong terms such as "structural relationship" and "decision-making logic" with more appropriate academic terms like "interrelationships," "associations," and "compositional patterns." We have clarified that the findings reveal co-occurrence patterns rather than establishing a hierarchical or causal structure. Please refer to the revised text in Lines 91-93, 101, 229-230, 624-644, 646-648, 651-652, and 748-750.

(3) The manuscript provides: Summary statistics, Reliability and validity outputs, Correlation matrices, Graphical representations of distributions. But it is not fully clear whether the raw SD scores (i.e., individual participant ratings for each church and each adjective pair) are provided in a reusable format (e.g., spreadsheet or table in Supporting Information). While not strictly required, best practice under PLOS policy encourages availability of data points underlying means and correlations. The authors should explicitly confirm that raw or semi-raw scoring matrices are included as Supporting Information, or provide them in a tabular format to enhance transparency, replicability, and secondary analysis.

Response: Thank you for your valuable and constructive suggestions. In strict accordance with PLOS data policy and the reviewer's suggestion, we have provided the complete raw scoring matrix and the aggregated mean scores as Supporting Information：

S1 Table： Aggregated mean scores of the six evaluation criteria for 62 churches. This table provides the summary metrics used for regional comparative analysis.

S2 Table： Raw individual scoring matrix for 62 churches. This dataset contains 3,100 individual data points (50 participants × 62 churches) used to calculate the reliability, validity, and mean values presented in the study.

Response to Reviewer 2:

The study addresses a topic that has historical, architectural, and religious significance. It is also relevant to the construction industry, particularly architecture, and has practical implications for cultural management departments and architectural heritage preservation. Additionally, the utilization of the semantic difference method to carry out the feature evaluation of the facades of the churches under study brings a novel dimension.

The manuscript exhibits several promising features, including its focus on the subject being investigated. The study contributes meaningfully to knowledge by addressing a methodological gap in earlier studies on comprehensive evaluations of facades of Catholic churches in the study area. Overall, the literature review presented in the literature review section is relevant and current to some extent. However, the following areas of the manuscript require attention to enhance its robustness.

(1)The introduction section of the study, though concise, presents an adequate background to the study. It could have benefited from more recent citations.

Response: Thank you for your valuable feedback. We appreciate your recognition of the background provided in the introduction. Following your suggestion, we have updated the references to reflect the most recent developments in this field. Specifically, we have incorporated six new citations from the last five years (e.g., Liao, 2021; Pynkyawati, 2022; Coomans, 2023; Yin et al., 2025; Wu et al., 2025; Zhang et al., 2025) to provide a more contemporary context for our study. Please refer to the revised text in Lines 43, 46-48,49,50-51,57-66, and 72.

(2)Also, there is no reference to literature between lines 70 and 75.

Response: We sincerely apologize for this oversight and thank the reviewer for pointing it out. In accordance with your suggestion, we have now included the necessary citations between lines 70 and 75 (in the revised manuscript). The added references (Coomans, 2023; Cody,1996; Arno, 2026), provide the required theoretical basis for this section. Please refer to the revised text in Lines 80, 82, and 84.

(3)Kindly rephrase the sentence between lines 83 and 85: "Furthermore, the relationship between facade structural characteristics is revealed from both the overall evaluation score of the church and the evaluation score of each individual church." The use of "the church" and "individual church" might not be clear to some audiences.

Response: Thank you for this insightful suggestion. We agree that the previous phrasing was somewhat ambiguous regarding the scale of the analysis. We have revised the sentence to distinguish more clearly between the entire sample set and the specific case studies. Please refer to the revised text in Lines 91-93.

(4)The sub-section, 3.1 (Research area) under the Research methodology section, provides the requisite information situate the study. However, the source of the literature and supporting Figure have not been provided.

Response: Thank you for your careful review and constructive feedback. We agree that the sources for the statistical data and Figure 1 should be more explicitly documented to ensure academic rigor. We have addressed these concerns in the revised Section 3.1 as follows:

Clarification of Data Sources: We have explicitly stated that the statistical data regarding the 62 Catholic churches were synthesized from the Digital Local Chronicles of Sichuan and Chongqing and historical records in Sichuan Catholicism (Liu, 2009). Additionally, we have incorporated more recent and relevant literature (e.g., Wang, 2020; Gao et al., 2025; Zhang, 2025) to provide a more robust theoretical and empirical background.

Documentation of Figure 1: We have updated the caption of Figure 1 to clarify its source. This figure was generated by the authors by integrating data from official local chronicles with our extensive field investigation findings.

Please refer to the revised text in Lines 199, 201, 202-204, and 212-213.

(5)Also, the limitations regarding the study participants are clearly stated in the discussion section, but the sampling/selection method has not been described in the methodology.

Response: Thank you for your valuable and constructive suggestions. We fully agree that a detailed description of the sampling and selection process is essential for ensuring methodological transparency. In the revised manuscript, we have expanded the description of the purposive sampling method and the specific inclusion criteria for participants. Specifically, we have clarified that the 50 participants were selected from senior undergraduate and graduate architecture programs. This criterion ensures that they possess the necessary professional background to provide reliable and informed ratings for the study. Please refer to the revised text in Lines 267-271.

(6)The study employs the semantic difference method to evaluate the main facade characteristics of these churches across five dimensions, namely: relationship with the environment, facade contour, facade style, facade decoration, and religious expression. It has six evaluation criteria, subdivided into facade style 1 and façade style 2. This should be explicitly stated for clarity.

Response: Thank you for your valuable and constructive suggestions. We appreciate the reviewer's suggestion regarding the clarity of our evaluation criteria. Accordingly, we have updated the Abstract, Methodology, and Conclusion to specify the five dimensions and six criteria. To clarify the distinction between the two style-related sub-criteria, we have now included their respective adjective pairs: "Chinese vs. Western" (Facade Style 1) for cultural origin, and "Folk vs. Official" (Facade Style 2) for architectural formality and social stratum. Please refer to the revised text in Lines 25-27, 242-243, and 724-727.

(7)Under subsection 4.4. (Correlation analysis of evaluation factors for façade) in the analysis section, kindly rephrase the sentence between lines 550 and 553 “Subsequently, correlation analysis will be performed on the calculated means of each church factor……” The portion “will be performed” should be “was performed”.

Response: Thank you for your meticulous review. We apologize for the inconsistent use of tenses in this section. Following your suggestion, we have changed "will be performed" to "was performed" to correctly reflect the completed nature of the data analysis. Furthermore, we have reviewed the entire subsection 4.4 and updated other future-tense expressions to the past tense for consistency. Please refer to the revised text in Lines 560 and 562.

(8) Also, could lines 558 and 597-598, which present the data’s “violation and significant departure from normalcy be explained further for clarity?

Response: Thank you for this insightful comment. We have expanded the explanation regarding the "departure from normalcy" in lines 558 and 597-598 to improve clarity.

Specifically, we have clarified that the normality test (K-S test) was a necessary step to justify our choice of statistical methods. Since the p-values were all below 0.05, the data exhibited a non-normal distribution, which statistically precluded the use of Pearson’s correlation. Therefore, we opted for Spearman’s rank correlation, a non-parametric approach that provides more reliable and accurate measures for data that do not follow a normal distribution. We have also refined the language, replacing "violated" with "exhibited a significant departure from" to align with standard academic reporting.Please refer to the revised text in Lines 567-569, 571-575, and 612-616.

(9)Please, rectify the following references: 786 Coomans, T., & Luo, W. (2015); 789 Coomans, T. (2025) and 832 Ocker, C., & Elm, S. (2020).

Response: Thank you for your careful check. In accordance with your suggestion, we have rectified the citations for 786 (Coomans & Luo, 2015), 789 (Coomans, 2025). Specifically, we have added missing information such as the names of the editors, publication locations, and full book titles to ensure they strictly follow the journal's formatting guidelines. Please refer to the updated References section (Lines 909-912, 913-914).

---

## [Decision Letter · Decision Letter 1]

21 Apr 2026

Comprehensive feature evaluation of the main facades of Catholic churches in Sichuan-Chongqing region based on semantic difference method, 1840-1949

PONE-D-25-67492R1

Dear Dr. Li,

We’re pleased to inform you that your manuscript has been judged scientifically suitable for publication and will be formally accepted for publication once it meets all outstanding technical requirements.

Kind regards,

Kofi Agyekum, Ph.D.

Academic Editor

PLOS One

Additional Editor Comments (optional):

Dear Author,

Thank you for submitting your revised manuscript to PLOS One. The reviewers have recommended it for publication. Congratulations! We hope you will consider

PLOS One again for your future manuscripts.

Reviewers' comments:

Reviewer's Responses to Questions

**Comments to the Author**

1. If the authors have adequately addressed your comments raised in a previous round of review and you feel that this manuscript is now acceptable for publication, you may indicate that here to bypass the “Comments to the Author” section, enter your conflict of interest statement in the “Confidential to Editor” section, and submit your "Accept" recommendation.

Reviewer #1: All comments have been addressed

Reviewer #2: All comments have been addressed

2. Is the manuscript technically sound, and do the data support the conclusions?

Reviewer #1: Yes

Reviewer #2: Yes

3. Has the statistical analysis been performed appropriately and rigorously?

Reviewer #1: Yes

Reviewer #2: Yes

4. Have the authors made all data underlying the findings in their manuscript fully available?

Reviewer #1: Yes

Reviewer #2: Yes

5. Is the manuscript presented in an intelligible fashion and written in standard English?

Reviewer #1: Yes

Reviewer #2: Yes

6. Review Comments to the Author

Reviewer #1: (No Response)

Reviewer #2: (No Response)

7. PLOS authors have the option to publish the peer review history of their article (what does this mean?). If published, this will include your full peer review and any attached files.

Reviewer #1: **Yes:** Henry Kofi Dansu

Reviewer #2: No

---

## [Editor Report · Acceptance letter]

PONE-D-25-67492R1

PLOS One

Dear Dr. Li,

I'm pleased to inform you that your manuscript has been deemed suitable for publication in PLOS One. Congratulations! Your manuscript is now being handed over to our production team.

Kind regards,

on behalf of

Prof. Kofi Agyekum

Academic Editor

PLOS One